# *Annona squamosa* Fruit Extract Ameliorates Lead Acetate-Induced Testicular Injury by Modulating JAK-1/STAT-3/SOCS-1 Signaling in Male Rats

**DOI:** 10.3390/ijms25105562

**Published:** 2024-05-20

**Authors:** Maysa A. Mobasher, Alaa Muqbil Alsirhani, Maha Abdullah Alwaili, Fadi Baakdah, Thamir M Eid, Fahad A. Alshanbari, Reem Yahya Alzahri, Sahar Abdulrahman Alkhodair, Karim Samy El-Said

**Affiliations:** 1Department of Pathology, Biochemistry Division, College of Medicine, Jouf University, Sakaka 72388, Saudi Arabia; mmobasher@ju.edu.sa; 2Department of Chemistry, College of Science, Jouf University, Sakaka 2014, Saudi Arabia; amassaf@ju.edu.sa; 3Department of Biology, College of Science, Princess Nourah bint Abdulrahman University, Riyadh 11671, Saudi Arabia; maalwaele@pnu.edu.sa; 4Department of Medical Laboratory Sciences, Faculty of Applied Medical Sciences, King Abdulaziz University, Jeddah 21589, Saudi Arabia; fbaakdah@kau.edu.sa; 5Special Infectious Agents Unit, King Fahd Medical Research Center, King Abdulaziz University, Jeddah 21589, Saudi Arabia; 6Department of Biochemistry, Faculty of Science, King Abdulaziz University, Jeddah 21589, Saudi Arabia; tmeid@kau.edu.sa (T.M.E.); salkhodair@kau.edu.sa (S.A.A.); 7Department of Medical Biosciences, College of Veterinary Medicine, Qassim University, Buraydah 51452, Saudi Arabia; shnbry@qu.edu.sa; 8Department of Biology, College of Science, University of Jeddah, Jeddah 21589, Saudi Arabia; ryalzahri@uj.edu.sa; 9Biochemistry Division, Chemistry Department, Faculty of Science, Tanta University, Tanta 31527, Egypt

**Keywords:** *Annona squamosa* fruit, antioxidant, anti-inflammatory, lead acetate, JAK-1/STAT-3/SOCS-1 axis

## Abstract

Lead (Pb) is a common pollutant that is not biodegradable and gravely endangers the environment and human health. *Annona squamosa* fruit has a wide range of medicinal uses owing to its phytochemical constituents. This study evaluated the effect of treatment with *A. squamosa* fruit extract (ASFE) on testicular toxicity induced in male rats by lead acetate. The metal-chelating capacity and phytochemical composition of ASFE were determined. The LD_50_ of ASFE was evaluated by probit analysis. Molecular docking simulations were performed using Auto Dock Vina. Forty male Sprague Dawley rats were equally divided into the following groups: Gp1, a negative control group; Gp2, given ASFE (350 mg/kg body weight (b. wt.)) (1/10 of LD_50_); Gp3, given lead acetate (PbAc) solution (100 mg/kg b. wt.); and Gp4, given PbAc as in Gp3 and ASFE as in Gp2. All treatments were given by oro-gastric intubation daily for 30 days. Body weight changes, spermatological parameters, reproductive hormone levels, oxidative stress parameters, and inflammatory biomarkers were evaluated, and molecular and histopathological investigations were performed. The results showed that ASFE had promising metal-chelating activity and phytochemical composition. The LD_50_ of ASFE was 3500 mg/kg b. wt. The docking analysis showed that quercetin demonstrated a high binding affinity for JAK-1 and STAT-3 proteins, and this could make it a more promising candidate for targeting the JAK-1/STAT-3 pathway than others. The rats given lead acetate had defective testicular tissues, with altered molecular, biochemical, and histological features, as well as impaired spermatological characteristics. Treatment with ASFE led to a significant mitigation of these dysfunctions and modulated the JAK-1/STAT-3/SOCS-1 axis in the rats.

## 1. Introduction

Global health is seriously threatened by industrial activities that significantly increase the quantity of heavy metals in the environment; pollution from these metals is becoming a problem of great concern due to its adverse effects around the world [1]. Impaired fecundity and infertility have long been issues, and they continue to be major concerns leading to significant clinical problems. Male infertility has been linked to various factors, including environmental, endocrinological, and genetic ones [2]. Environmental exposure to heavy metals negatively affects male sperm production and fertility by aggravating oxidative stress [3].

One significant risk factor influencing male fertility is the toxicity of heavy metals such as lead encountered in the workplace [4]. Lead is a hazardous heavy metal that is widely dispersed in the environment. Because of its physicochemical characteristics, non-biodegradability, and continuous use, it accumulates in the environment, posing a growing risk to living organisms [5]. Lead is a systemic toxicant, affecting almost all organ systems via interference with calcium’s regulatory effect on cell functioning and disrupting various intracellular biological processes. It can disrupt the neurological, reproductive, hematopoietic, and cardiovascular systems [6]. Furthermore, lead exposure generates oxidative stress in biological systems that deactivates enzyme functioning, impairs immunological function, and damages the architecture of tissues [7]. Reduced sperm motility and quantity, increased aberrant morphology, altered spermatogenesis, chromosomal damage, improper prostatic function, altered levels of testosterone, and infertility were among the reported repro-toxic effects of lead exposure [8]. The biochemical compositions and histopathological structures of the testis were altered by lead intoxication, and this affected steroidogenesis in experimental animals [9]. It has been reported that the seminiferous tubules of rats exposed to lead acetate were notably atrophied due to oxidative stress, which disrupted the maturation of spermatozoa [10].

Oxidative stress, a process that is directly implicated in the pathophysiology of numerous disorders, can be alleviated by plant-derived phytochemicals [11,12]. Plants are sources of natural ingredients that are widely used as medicines. Natural antioxidants play a critical preventive role in testicular toxicity and could be a successful treatment for testicular heavy metal toxicity [13]. *Annona squamosa* (custard apple), belonging to the Annonaceae family, is found in tropical and subtropical regions and has various traditional uses that indicate it might be applicable for medical purposes [14]. Phytochemical investigations of *Annona* have identified several of its bioactive metabolites such as flavonoids, terpenoids, coumarins, anthraquinones, and phytosterols [15]. *A. squamosa* has been reported to have anticancer, antioxidant, anti-inflammatory, antidiabetic, antihypertensive, hepatoprotective, and antiparasitic activities [16]. *A. squamosa* extracts exhibit potent hepatoprotective activities against alcohol-induced liver injury in rats because of their antioxidant benefits [17]. Moreover, *A. squamosa* extract has been reported to have anti-hyperglycemic and ameliorative effects on the hepato-renal dysfunction associated with diabetes mellitus in rats [18]. A previous study indicated that *A. squamosa* had broad-spectrum activity against bacterial pathogens, which suggested that it could be used as a food sanitizer [19]. Leaf extract from another member of the Annonaceae family, *A. muricata*, has been studied for its ability to shield male rats from testicular damage caused by cadmium [20]. It has been reported that *A. muricata* attenuated testicular oxidative stress in rats [21].

Inflammatory cytokines, including interleukin-6 (IL-6), regulate testicular steroidogenesis and spermatogenesis [22]. The intracellular signal transduction in response to these inflammatory cytokines occurs via the Janus kinase (JAK)-signal transducer and activator of transcription (STAT) signaling pathway [23]. Cytokine signaling via the JAK/STAT pathway is negatively inhibited by the suppressor of cytokine signaling (SOCS) proteins. Of the eight SOCS proteins, SOCS-1 is the most effective and can directly inhibit JAK/STAT transcription [24]. IL-6 triggers the JAK-associated receptor, accelerating tissue fibrosis via activating STAT-3 to initiate downstream signals [25]. It has been reported that IL-6 could inhibit the differentiation of rats’ stem Leydig cells [26]. Alves Silva et al. (2021) suggested that a lack of IL-6 drives androgenic production by increasing SOCS levels in the testis, leading to an increase in spermatogenesis [27]. Furthermore, a previous study reported the anti-inflammatory effects of natural products against testicular inflammation via the modulation of JAK/STAT signaling in rats [28]. Prior reports documented the beneficial effects of medicinal plants against testicular damage caused by lead intoxication [29,30]. Indeed, *A. squamosa* fruit extract (ASFE) had several therapeutic properties that could be used to control physiological and biochemical deregulations. However, the role of ASFE as a natural chelating agent that could mitigate testicular damage triggered by lead has not been previously investigated. Therefore, this novel study investigated the effect of ASFE on mitigating the testicular damage induced by lead acetate in male rats.

## 2. Results

### 2.1. Phytochemical Constituents of ASFE

The results showed that the *A. squamosa* fruit (ASF) yielded an adequate amount (10.5%) of extract. The metal-chelating activity of ASF was 82% ± 3.87, and the half-maximal effective concentration (EC_50_) was 383.65 ± 4.95 μg/mL. The total phenolic content of ASF was 25.48 ± 2.56 mg GAE/g DW. The total flavonoid content was 16.75 ± 1.87 mg QE/g DW. The level of saponin was 342 ± 3.97 mg/g DW. The total antioxidant capacity (TAC) of the fruit extract was 67.78 ± 3.58 mg AAE/g DW, and the scavenging activity percentage of the 2,2-diphenyl-1-picrylhydrazyl (DPPH) (%) was 86% ± 4.02. The amount of ASFE that could reduce 50% of the DPPH was 5.81 ± 0.80 mg/mL (Table 1).

### 2.2. Gas Chromatography–Mass Spectrometry (GC-MS) Analysis of ASFE

The GC-MS analysis showed that ASFE contains several chemical constituents. The abundant phytoconstituents detected in ASFE were 4H-pyran-4-one, 2,3-dihydro-3,5-dihydroxy-6-methyl; butylated hydroxytoluene; 5,8,11-heptadecatrienoic acid, methyl ester; phytol; eicosanoic acid; oxiraneundecanoic acid, 3-pentyl-, methyl ester; β-sitosterol; hexadecanoic acid, ethyl ester; octadecanoic acid; and quercetin. Their retention times were 2.25, 8.73, 18.59, 19.35, 24.43, 25.76, 27.68, 39.72, 43.58, and 45.63 min, respectively. The peak area percentages were 6.17%, 4.80%, 8.61%, 5.36%, 12.51%, 6.42%, 7.86%, 12.36%, 5.74%, and 7.85%, respectively (Figure 1 and Table 2).

### 2.3. Bioactive Compound Interactions with JAK-1, STAT-3, and SOCS-1 Proteins by Molecular Docking

Table 3 shows the ∆G (kcal/mol) and binding affinity values for the JAK-1, STAT-3, and SOCS-1 proteins with the bioactive compounds in the ASFE. Of the docked compounds, the quercetin derivative exhibited the highest binding affinity towards JAK-1 (−9.1 kcal/mol), followed by β-sitosterol (−7.9 kcal/mol) and Bis(2-ethylhexyl) phthalate (−6.3 kcal/mol). In the case of STAT-3, the top-scoring compounds were the quercetin derivative (−7.9 kcal/mol), β-sitosterol (−6.8 kcal/mol), and caryophyllene (−6.4 kcal/mol). For SOCS-1, β-sitosterol (−7.1 kcal/mol), butylated hydroxytoluene (−6.2 kcal/mol), and caryophyllene (−6.5 kcal/mol) displayed the strongest binding affinities.

Figure 2 provides information on the best compound interactions with the three proteins in both 2D and 3D forms. Figure 2 shows the interaction of the quercetin derivative with JAK-1, with various types of hydrogen bonds and other interactions mentioned. Figure 2 describes the interaction of the quercetin derivative with STAT-3, featuring conventional hydrogen bonds, a carbon–hydrogen bond, and pi–alkyl and pi–sigma bonds. Figure 2 highlights the interaction of β-sitosterol with SOCS-1, which primarily involves pi–alkyl and pi–sigma bonds. Overall, these compound–protein interactions suggest a potential for binding and affinity between the compounds and proteins, which may have implications for drug development or therapeutic interventions.

### 2.4. The LD_50_ of ASFE in Rats

The LD_50_ was determined after 24 h of oral treatment using different doses of ASFE in rats. Different groups were injected by gavage with different doses ranging from 1 to 5 g/kg. The results of probit analysis showed that the LD_50_ of ASFE was 3500 mg/kg. The animals showed no stereotypical toxic symptoms such as convulsion, ataxia, diarrhea, or increased diuresis except at 3500 mg/kg (see Figure 3).

### 2.5. Effect of ASFE Treatment on Body Weight, Epididymis, and Testis Weights of Rats Given Lead Acetate

There were no significant changes in the percentages of body weight (% b. wt.) in the normal control and ASFE control groups (30.26 and 34.75%, respectively). However, rats that received PbAc showed a significant decrease (*p* < 0.05) in their % b. wt. change (11.58%) compared with the control groups. Treatment of PbAc-intoxicated rats with ASFE led to a significant improvement (*p* < 0.05) in the % b. wt. to 22.98% when compared with the group given PbAc (Figure 4A). The testis and epididymis weights were significantly decreased (*p* < 0.05) in the PbAc-administered group when compared to the control groups by 1.7- and 1.5-fold, respectively. However, the PbAc/ASFE group showed a significant improvement in the testis and epididymis weights when compared to the PbAc-administered group (Figure 4B).

### 2.6. Effect of ASFE Treatment on Lead Residues in Testis of PbAc-Intoxicated Rats

The effect of ASFE administration on lead residues in the testis of male rats that had received lead acetate was evaluated. The data showed that the residual lead in the testis was significantly increased (*p* < 0.05) in the group given PbAc compared to the control group. However, treatment of the PbAc-intoxicated group with ASFE led to a significant reduction in testicular lead residue when compared to the PbAc-administered group alone (Figure 5).

### 2.7. Effect of ASFE Treatment on Spermatological Parameters of PbAc-Intoxicated Rats

Spermatozoa morphological investigations showed that the anomalies in the head portions were more noticeable than those in the tail and intermediate sections. Compared with the control group (11.82 ± 1.55%), the highest abnormality percentage was recorded in the PbAc-intoxicated group (19.62 ± 2.59%). However, the group given lead acetate had a lower percentage of sperm abnormalities (13.54 ± 2.89%) (Table 4). In the PbAc-administered group, sperm count was significantly decreased (*p* < 0.01) versus the control group (31.77 ± 2.79 × 10^6^/mL versus 68.91 ± 3.87 × 10^6^/mL). The treatment of the PbAc-injected group with ASFE LD_50_ resulted in a significant increase in the sperm count to nearly that of the control groups (55.69 ± 3.48 × 10^6^/mL) compared with the PbAc-injected group (Table 4). Lead acetate administration reduced the sperm viability percentage in rats by up to 27.82% ± 2.95%. In the group given lead acetate and then ASFE, the percentage of sperm viability increased close to that of the control groups (50.75% ± 3.86). Furthermore, a significant decrease (*p* < 0.01) in the motility percentage of the PbAc-intoxicated group was recorded (28.21 ± 3.15%). It was determined that the sperm motility of the group given lead acetate and then ASFE was not different from that of the normal control group (Table 4).

### 2.8. Effect of ASFE on Levels of Reproductive Hormonal in Male Rats Given Lead Acetate

Lead intoxication in male rats resulted in a significant reduction (*p* < 0.05) in levels of luteinizing hormone (LH), at 2.03 ± 0.15 ng/mL; follicle-stimulating hormone (FSH), at 3.48 ± 0.21 ng/mL; and testosterone, at 1.42 ± 0.13 ng/mL, compared with the control groups. As compared to the PbAc-administered group alone, the treatment of PbAc-intoxicated rats with ASFE led to significant improvements (*p* < 0.05) in levels of LH (2.96 ± 0.19 ng/mL), FSH (4.68 ± 0.25 ng/mL), and testosterone (2.01 ± 0.18 ng/mL) (Figure 6).

### 2.9. Treatment of PbAc-Intoxicated Rats with ASFE Alleviated Oxidative Stress on Testicular Tissues

The levels of malondialdehyde (MDA) were markedly increased in the PbAc-injected group (44.67 ± 3.86 nmol/g tissue) compared with the negative control group (21.73 ± 1.95 nmol/g tissue) and the control group given AFSE (17.92 ± 1.45 nmol/g tissue). Treatment with 1/10 of ASFE LD_50_ in the PbAc-intoxicated group led to a significant decrease (*p* < 0.05) in the testicular MDA level, to nearly that of the control groups. Moreover, the treatment of PbAc-intoxicated rats with ASFE led to a marked decrease (*p* < 0.05) in testicular protein carbonyl levels compared with the lead acetate-intoxicated rats alone (Table 5). In contrast, the superoxide dismutase (SOD), catalase (CAT), and glutathione (GSH) levels were significantly decreased (*p* < 0.05) in the PbAc-intoxicated rats compared with the control groups. The PbAc/ASFE-treated group had a significant improvement in this reduction in antioxidant biomarkers (see Table 5).

### 2.10. Effects of ASFE Treatment on Inflammatory Cytokines of PbAc-Intoxicated Rats

The results showed that there were no significant alterations in inflammatory cytokines between the normal control group and the ASFE control groups. In the rats with testicular injury induced by lead acetate, the levels of inflammatory biomarkers, including interleukin-6 (IL-6), tumor necrosis factor alpha (TNF-α), nuclear factor kappa B (NF-κB), and cyclooxygenase-2 (COX-2), were significantly elevated (*p* < 0.01) in terms of testicular supernatants compared with the negative control and ASFE control groups. However, these inflammatory cytokines were significantly decreased (*p* < 0.01) in the PbAc/ASFE-administered group compared with the group of lead acetate-intoxicated rats (Table 6).

### 2.11. Treatment with ASFE Mitigated Inflammation by Modulating JAK-1/STAT-3/SOCS-1 Signaling in PbAc-Intoxicated Rats

In rats with testicular injury caused by lead acetate, the mRNA expression levels of *JAK-1* and *STAT-3* were markedly increased. However, ASFE treatment significantly reduced the expression levels (*p* < 0.05). Gene expression of *SOCS-1* was significantly reduced in the testis homogenate of rats given lead acetate compared with the control group (*p* < 0.05), but treatment of these rats with ASFE greatly restored *SOCS-1* gene levels to nearly those of the control group (Figure 7 and Figure 8). Testicular expression of the *IL-6*, *TNF-α*, *NF-κB*, and *COX-2* genes was significantly increased (*p* < 0.05) in the PbAc-intoxicated groups, but treatment with ASFE led to significant reductions in these levels (Figure 7 and Figure 8).

### 2.12. Treatment with ASFE Significantly Improved Testicular Histopathological Damage Induced by PbAc in Rats

Microscopic examination of testicular tissue from the control group and the ASFE control group showed a normal histological structure of the active, mature, functioning seminiferous tubules and Sertoli cells and the presence of active sperms in the lumens as well as Leydig cells in the interstitial tissues (see Figure 9A–D). In contrast, testicular sections of the PbAc-challenged rats showed marked alterations of the testicular tissue and the degeneration of most seminiferous tubules with an absence of spermatogenic series; congestion of the testicular vessels; an irregular, damaged basement membrane; and severe degenerative changes of the spermatogonial cells. Leydig cell degeneration, scattered necrosis, and Sertoli cells with large spermatid giant cells in their lumens were observed along with dispersed seminiferous tubules (see Figure 9E–H). Interestingly, the testis of the rats given lead acetate and then treated with ASFE had a restored architecture of most of the seminiferous tubules, improved testicular histology, and improved spermatogonial cell integrity, as well as a markedly restored process of spermatogenesis with normal-appearing Leydig cells (see Figure 9I,J).

The histological structure and spermatogenesis in rats were evaluated by the Johnsen scoring system. The seminiferous tubules were graded (1 to 10) according to the reduction in the number and density of germ cells from the lumen of seminiferous tubules (Table 8). Scores in the seminiferous tubule cross-sections of control and ASFE control groups were 10, indicating spermatogenesis with several spermatozoa present in a section of seminiferous tubules (Figure 9A–D). Scores 4–6 indicated that seminiferous tubules of PbAc-intoxicated rats showed no spermatozoa, but spermatids and germ cells were present (Figure 9E–H). Treatment with PbAc/ASFE showed an increase in the score for the analysis of testicular tissues to 9 (Figure 9I,J).

## 3. Discussion

The pollution of natural resources such as water, soil, and air is caused by the growing usage of fossil fuels and the discharge of industrial waste. Exposure to heavy metals poses significant health and environmental concerns [31]. The decontamination of heavy metals has been efficiently performed using hydrogels, which demonstrated exceptional adsorption of heavy metal ions [32,33]. Among workers exposed to lead, male infertility was induced by the alteration of the hypothalamic–pituitary axis via the development of reactive oxygen species (ROS), which in turn impairs spermatogenesis [34]. Administering chelating drugs to form an insoluble complex with lead (Pb) is the approved treatment for lead toxicity. Unfortunately, most of these drugs have negative side effects, including causing the redistribution of toxic metals to other unaffected tissues or the loss of vital metals [35]. Therefore, developing safe, natural, and affordable agents to combat Pb poisoning is crucial. The ethnopharmacological approach, which is based on the traditional medical system, is the foundation for the development of phytopharmaceutical medications [9,36]. Phytochemical studies determined the bioactive principles from extracts from Annona that have antioxidant properties and have several biomedical applications [16]. The traditional uses, phytochemistry, and pharmacological activities of Annona have been reviewed [37]. *A. squamosa* is a deciduous tropical evergreen fruit that has recently attracted the attention of scientists for its medicinal value [38]. The purpose of this investigation was to ascertain the mechanisms by which ASFE treatment mitigated the testicular injuries inflicted in rats by lead acetate.

The present study showed that ASFE had promising quantities of phytochemical constituents, and this agreed with previous reports, which demonstrated that various phytochemicals in *A. squamosa* had pharmacological and therapeutic impacts [37,39]. In this study, the metal-chelating activity of ASFE was in congruence with the previous study by Loizzo et al. (2012), who reported the antioxidant and metal-chelating activities of Annona that prevented oxidative stress-mediated pathogenesis. Moreover, the DPPH scavenging capacity of ASF was 86%, and its IC_50_ was 5.81 mg/mL [40,41]. A previous study evaluated the antioxidant activity of Annona and reported the free radical scavenging properties of *A. squamosa* [39]. In the current study, a GC-MS investigation screened the bioactive compounds of ASFE and found that potent phytochemical compounds were present. A previous report indicated the presence of bioactive secondary metabolites in Annona extracts; these plant-derived metabolites could be used for drugs and detain the breakthrough of various diseases [42]. Interestingly, the most abundant phytochemical compounds detected in ASFE had the highest peak and the ability to enhance a variety of biological functions. 4H-pyran-4-one, 2,3-dihydro-3,5-dihydroxy-6-methyl exists in various natural extracts and foods, and its potent antioxidant activity was reported by many researchers [43]. Butylated hydroxytoluene is used as a chemical preservative in food, drugs, and other related products. There is a widespread use of antioxidants with great economic advantages. Furthermore, the substance could be useful for various anti-inflammatory formulations [44]. One of the promising phytoconstituents detected in ASFE in the present investigation was phytol, a diterpene with a wide range of applications in the pharmaceutical industry, including anti-inflammatory, immune-modulating, anxiolytic, metabolism-modulating, antioxidant, and antimicrobial effects [45]. Eicosanoic acid, hexadecanoic acid, and octadecanoic acid are key bioactive fatty acids with various potential therapeutic applications. β-sitosterol, an essential major phytosterol that was detected in ASFE, has potential therapeutic properties for various types of cancer and other biomedical properties [46]. Quercetin, a flavonoid molecule, could chelate metal ions when DNA damage occurs and prevent different oxidative radicals from developing [47,48].

It is noteworthy that several bioactive compounds in ASFE exhibited comparable binding affinities across JAK-1, STAT-3, and SOCS-1 target proteins, suggesting that they may have multi-target activities. The quercetin derivative, β-sitosterol, and caryophyllene demonstrated relatively high binding affinities for JAK-1, STAT-3, and SOCS-1 proteins, implying that they have potential as multi-target inhibitors or modulators. The quercetin derivative demonstrated a strong binding affinity (−9.1 kcal/mol) towards JAK-1, suggesting a potentially potent interaction. This high affinity can be attributed to the presence of diverse intermolecular interactions, including multiple conventional hydrogen bonds with key residues (GLU83, ARG108, TYR412, MET109, TYR112, and TYR281). These hydrogen bonds create highly directional and strong attractions, promoting a stable complex. Additionally, a pi–pi T-shaped interaction with TYR281, involving aromatic ring stacking, further contributed to complex stability. The presence of a pi–anion interaction with GLU284 leveraged the electrostatic attraction between the ligand and a negatively charged residue, enhancing the binding affinity. Interestingly, the docking analysis identified an unfavorable bump with ARG110. This steric clash highlighted the importance of optimal ligand conformation for maximizing favorable interactions. In contrast, the β-sitosterol ligand exhibited a moderate binding affinity (−7.1 kcal/mol) towards SOCS-1. This lower affinity can be explained by the limited number of interactions observed confirming the upregulation of *SOCS-1* gene expression in this study. The primary interactions identified were pi–alkyl interactions with PHE56, involving weak attractions between aromatic rings and aliphatic chains. Additionally, a pi–sigma interaction with THR65 provided a minimal contribution to binding. The quercetin derivative again displayed a substantial binding affinity (−7.9 kcal/mol) towards STAT-3; this strong affinity was likely due to a combination of interactions. Conventional hydrogen bonds with CYS712 and GLU638 play a significant role in binding strength and complex stability. A carbon–hydrogen bond with VAL637 was observed, but its contribution was likely weak. The ligand also participated in a pi–alkyl interaction with PRO728, offering some stabilization. The number and type of intermolecular interactions significantly influence binding affinity. The stronger binding affinities observed for JAK-1 and STAT-3 compared with *SOCS-1* can be attributed to the presence of multiple hydrogen bonds and aromatic stacking interactions with the quercetin derivative. These findings suggested that the quercetin derivative may be a more promising candidate for targeting the JAK-1/STAT-3 pathway due to its superior binding affinities compared with β-sitosterol with *SOCS-1*. Understanding these intermolecular interactions is crucial for drug discovery. By optimizing the ligand structure to promote favorable interactions with target proteins, researchers can develop more potent and selective drugs with improved therapeutic potential.

The efficacy and safety of using ASFE in rats were evaluated by determining the acute oral toxicity of the ASFE. The results revealed that treated rats showed no stereotypical toxic symptoms such as convulsion, ataxia, diarrhea, or increased diuresis except at 3500 mg/kg. Therefore, in our study, we used 1/10 of this LD_50_ value in the treatment protocol. A previous study demonstrated that the median lethal dose of Annona senegalensis was found to be more than 3000 mg/kg in rats [49]. Furthermore, it has been reported that the LD_50_ of the *A. squamosa* leaves was 5000 mg/kg body weight, indicating that the extract was well tolerated [50]. Treatment with ASFE following lead acetate intoxication led to a significant improvement in the % b. wt of rats. These findings agreed with previous investigations reporting weight loss and reduced intestinal absorption upon exposure to lead in experimental animals [51]. A previous study reported the improvement of body weight loss in rats that was induced by lead upon treatment with natural products. This may result from the removal of the harmful effects of lead from the rats’ tissues and circulation, enhancing nutrition and metabolism, and it indicates that ASFE may have a mitigating effect on lead acetate toxicity. This finding was consistent with previous studies showing the impact of plant extracts on body weight loss brought about by lead toxicity [52,53]. Furthermore, the epididymis and testis weights were significantly decreased due to the injection of PbAc. Treatment of the PbAc-administered group with ASFE led to significant improvement in the epididymis and testis weights [9]. The current investigation demonstrated that the residual lead in the testis of the PbAc-administered group was the highest. This may be due to the accumulation of lead heavy metals in the testis after the heavy metals had penetrated the blood–testis barrier. These results agreed with Marchlewicz et al. (2009) [54]. However, treatment of the PbAc-intoxicated group with ASFE led to a significant reduction in the testicular lead residue. This finding agreed with that of a previous study that reported the effect of lead toxicity on male rats’ reproductive function with the ameliorative role of natural products [9,52].

In the current investigation, the data clearly showed that the PbAc/ASFE administered group had a lower percentage of sperm abnormalities and a significant restoration of sperm count. This was in accordance with previous studies investigating the impact of plant-derived natural products on some reproductive metrics and the improvement of lead acetate-induced testicular injury [9,30,55,56]. There are two ways that lead can poison the testis: directly, by attacking the Sertoli cells in the testicles, or indirectly, by affecting the hypothalamic–pituitary–testicular axis, which lowers the amount of testosterone, FSH, and LH [57]. The PbAc-intoxicated rats that were treated with ASFE showed significant improvement in the LH, FSH, and testosterone levels as compared with the rats given lead acetate but not ASFE. This could be due to the presence of promising phytocompounds in ASFE that affect the hypothalamic–pituitary axis and thus restore these hormones. These findings suggest the ameliorative effect of ASFE against testicular toxicity induced by lead acetate and were consistent with previous studies [57,58,59,60].

Exposure to heavy metals alters the antioxidant defenses, causes an imbalance in radical generating and scavenging activities, and generates oxidative products [61]. Oxidative stress directly decreases the quality of sperm and exerts harmful effects on testicular tissues through ROS generation. Plant-derived antioxidants could be a useful therapy for heavy metal toxicity in the testis [58]. The present study reported significant increases in the testicular MDA and protein carbonyl levels along with a significant decrease in the testicular SOD, CAT, and GSH levels of the PbAc-injected group. This indicated the breakdown of antioxidant defense mechanisms and the lipid peroxidative degradation of the bio-membrane. The treatment with ASFE led to a notable improvement in the antioxidant/oxidant status of the PbAc-intoxicated rats. These findings supported the potential antioxidant power of ASFE, which may be attributed to its cytoprotective properties, which make it capable of shielding testicular tissues against the oxidative stress induced by PbAc in rat testes. These data, in line with previous reports, demonstrated the promising role of naturally occurring ingredients in mitigating the oxidative stress induced by lead toxicity [60,62]. Oxidative stress causes a cascade of cellular signals that set off several inflammatory processes. It is among the core causes of male infertility, and it can induce levels of proinflammatory mediators that are higher than usual in the male reproductive tract [63]. Under physiological conditions, inflammatory mediators such as TNF-α, IL-6, COX-2, and the transcription factor NF-κB have crucial regulatory roles in spermatogenesis [64]. IL-6 disrupts the integrity of the blood–testis barrier by altering membrane proteins; elevated IL-6 levels during inflammation may impede testosterone synthesis and alter spermatogenesis. In this study, treatment with ASFE alleviated testicular inflammation that was promoted by the injection of lead acetate in rats. This was evidenced by a significant reduction in inflammatory biomarkers in testis tissues, which suggests the anti-inflammatory properties of ASFE against PbAc-induced testicular inflammation. These findings agreed with previous studies, which determined that natural products could lower levels of the previously mentioned inflammatory mediators in testicular tissues, thereby reducing inflammation [63,64,65,66].

In initiating the JAK-STAT signaling pathway, inflammatory cytokines bind to their receptors, leading to their activation and triggering the transcription of genes [67]. In this study, a significant increase in the testicular JAK-1 and STAT-3 genes was recorded in rats given lead acetate. This could be due to the significant production of inflammatory cytokines, which initiated the JAK-STAT signaling, that was confirmed in our investigation. This also caused a significant increase in the expression levels of inflammatory mediators (i.e., NF-κB and COX-2), a finding that agreed with the study of Nunes et al. (2017) [68]. ASFE treatment significantly reduced the testicular JAK-1 and STAT-3 gene expression of PbAc-challenged rats, suggesting the anti-inflammatory potential of ASFE. It is possible that ASFE suppressed the JAK-STAT signaling pathway by reducing the inflammatory biomarkers in this study. The SOCS proteins have vital roles in the regulation of inflammatory cascades and JAK-STAT signaling [69]. The mechanism of action of SOCS-1 and its potential therapeutic role in the prevention and treatment of autoimmune diseases have been reported [70]. STAT-3 and SOCS-1 are related to both IL-6 signaling and testicular functions [71]. In the present research, the relative gene expression of SOCS-1 was significantly reduced in the PbAc-treated rats, and ASFE restored the SOCS-1 gene to levels close to the control group. A previous study reported the effect of natural products on testicular inflammation via SOCS-1 modulation in rats [28]. Adeyi et al. (2023) investigated the beneficial role of ferulic acid as an anti-inflammatory agent via the modulation of SOCS-1 signaling pathways in testicular inflammation in a rat model [66]. Lead acetate exposure caused significant histopathological alterations in the testis architectures that were consistent with previous studies [51,56,72]. However, in this investigation, treatment with ASFE for lead acetate intoxication improved sperm characteristics significantly and affirmed the normal histological structure of most seminiferous tubules. This could be due to the various antioxidant effects of ASFE that corrected the negative impacts of various oxidants induced by PbAc. Previous studies demonstrated the impact of natural plant extracts on testicular histological injury induced by lead toxicity, and they supported our findings [9,56,73].

## 4. Materials and Methods

### 4.1. Chemicals

Lead acetate (PbAc) (Cat. No. 6080-56-4) and a rat tumor necrosis factor alpha (TNF-α) ELISA kit were purchased from Sigma-Aldrich (St. Louis, MO, USA). A rat testosterone ELISA kit was purchased from Crystal Chem Company, Elk Grove Village, IL, USA. Rat ELISA kits for follicle-stimulating hormone (FSH), protein carbonyl (PC), nuclear factor kappa-B (NF-κB), and cyclooxygenase-2 (COX-2) were purchased from MYBIOSOURCE Company (San Diego, CA, USA). Rat luteinizing hormone (LH) and interleukin 6 (IL-6) ELISA kits were purchased from Elabscience Company (Houston, TX, USA). Malondialdehyde (MDA), superoxide dismutase (SOD), catalase (CAT), and reduced glutathione (GSH) kits were purchased from Bio-diagnostic Company, Cairo, Egypt. IL-6 polyclonal antibodies were purchased from Thermo Fisher Scientific (Waltham, MA, USA). Diaminobenzidine (98%) was purchased from Sigma-Aldrich (St. Louis, MO, USA).

### 4.2. Collection and Preparation of Plant Materials

*Annona squamosa* fruits were obtained from the Crop Institute Agricultural Research Centre in Giza, Egypt. The plant met all applicable institutional guidelines and was authenticated by a specialist who followed the international guidelines and legislation. Chopped fruits were ground into a powder after being dried in the shade. After the filtration of 500 mL of 70% ethanol and 50 g of powdered fruits, *A. squamosa* fruit extract (ASFE) was dried [12].

### 4.3. Phytochemical Analysis of ASFE

The metal chelation power of ASFE was determined [74]. Total phenolic and flavonoid contents, total antioxidant capacity, saponin, and anthocyanin were evaluated in ASFE [75,76,77]. Furthermore, a spectrophotometric assessment was used for DPPH radical scavenging capability [78].

### 4.4. Gas Chromatography and Mass Spectrum (GC-MS) Profiling of ASFE

Using the Trace GC 1310-ISQ mass spectrometer, or “GC-MS” (Thermo Scientific, Austin, TX, USA), phytochemicals of ASFE were identified. The components were identified by matching their mass spectra and retention periods to those found in the mass spectral databases of WILEY 09 and NIST 11.

### 4.5. Molecular Docking Analysis

The protein structures for JAK-1 (UniProt ID: A0A8I6AJU1), STAT-3 (UniProt ID: P52631), and SOCS-1 (UniProt ID: Q9QX78) were obtained. The JAK-1 structure was modeled using the Robetta server and further refined using the ModRefiner tool. The STAT-3 structure was retrieved from the Protein Data Bank, while the SOCS-1 structure was obtained from the UniProt database. The active sites of these proteins were predicted using the DeepSite server, which employs deep learning algorithms to identify potential binding sites [79]. The prepared protein structures were further processed using AutoDock Tools 1.5.7. The resulting PDBQT files were used as input for molecular docking simulations. The ligand molecules were retrieved from the PubChem database in their SDF formats. These ligands were then minimized using the Avogadro 1.2.0 software, employing the Force Field (MMFF94) and the Conjugate Gradients algorithm. Molecular docking simulations were performed using AutoDock Vina [80]. The search space for the docking simulations was defined based on the predicted active site regions obtained from the deep site server. The docking results were visualized and analyzed using the BIOVIA Discovery Studio 2020 (San Diego, CA, USA).

### 4.6. Determination of the Median Lethal Dose (LD_50_) of ASFE

Twenty male Sprague Dawley rats (150 ± 5 g) were divided into 5 groups (*n* = 4) to estimate the LD_50_ following oral administration of ASFE. These groups were injected with ASFE (1–5 g/kg b. wt.). Rats were then observed for 24 h to see if there were any signs of toxicity and to assess the LD_50_. The LD_50_ value was determined by the probit analysis [81].

### 4.7. Animals and Experimental Design

Forty adult male Sprague Dawley rats (150  ±  5 g) were purchased from the National Research Center (NRC, Cairo, Egypt). This study was carried out according to the Research Ethical Committee guidelines of the Faculty of Science, Tanta University, Egypt (IACUC-SCI-TU-0294). Rats were equally divided into four groups: The first group (Gp1) was a negative control group. Gp2 was orally administered ASFE (350 mg/kg bwt) (1/10 of LD_50_). Gp3 was orally administered lead acetate (PbAc) solution (100 mg/kg bwt) [82]. Gp4 was orally administered ASFE as in Gp2 and with PbAc as in Gp3 (Figure 10). The percentages of body weight (% b. wt.) changes were determined. Rats were sacrificed under anesthesia (isoflurane), and blood samples were withdrawn from the rats intracardially for sera collection. Testis and epididymis tissues were excised, and their weights were recorded. The sperm abnormalities, count, viability, and motility were assessed. Parts of rat testicular tissue were used for molecular and biochemical analysis. The lead residue was determined in testicular homogenate after different treatments by using the PerkinElmer 2380 atomic absorption spectrophotometer. Furthermore, testicular sections were kept in 10% buffered formaldehyde for histopathological investigations.

### 4.8. Semen Analysis

To assess sperm motility, diluted epididymal sperm suspensions from different groups were transferred to a hemocytometer and examined under a light microscope (400×). The vitality of the sperm was assessed using the eosin stain by dye exclusion test, which involved staining two drops of eosin solution and one drop of collected semen. Spermatozoa that are viable cannot absorb the dye, but nonviable spermatozoa can. Sperm drops were stained with Wells and Awa’s stain and examined for abnormalities [83].

### 4.9. Biochemical Analysis

Luteinizing hormone (LH) (Cat. No. E-EL-R0026), follicle-stimulating hormone (FSH) (Cat. No. MBS2021901), and testosterone levels (Cat. No. 80550) were determined in serum using their ELISA kits. According to the manufacturer’s instructions, MDA (Cat. No. MD2529), PC (Cat. No. MBS2600784), SOD (Cat. No. SD2521), CAT (Cat. No. CA2517), and GSH (Cat. No. GR2511) were assessed in the testicular supernatant. Furthermore, rats’ ELISA kits were used to determine inflammatory biomarkers, including interleukin-6 (IL-6) (Cat. No. E-HSEL-R0004), tumor necrosis factor-α (TNF-α) (Cat. No. RAB0479), nuclear factor kappa-B (NF-κB) (Cat. No. MBS453975), and cyclooxygenase-2 (COX-2) (Cat. No. MBS266603) in the testicular homogenate.

### 4.10. Molecular Analysis

The mRNA expression of *JAK-1*, *STAT-3*, *SOCS-1*, *IL-6*, *TNF-α*, *NF-κB*, and *COX-2* genes was evaluated in the testicular tissues by SYBR Green. Using the β-actin gene as an internal reference, the primers were prepared using the Primer-Blast program from NCBI (Table 7). The relative expression of target genes in testicular tissues was calculated by the method of Livak and Schmittgen (2001) [84].

### 4.11. Histopathological Investigations

Testis tissues were cut and fixed in buffered formalin (10%), dehydrated, cleaned in xylene, embedded in paraffin wax, and sectioned at 5 μm. The sections were stained with hematoxylin and eosin (H & E), examined under a light microscope (Olympus CX31, Tokyo, Japan), and photographed with a digital camera (Olympus Camedia 5060, Tokyo, Japan) [85]. Johnsen’s mean testicular score was used to evaluate testicular damage and spermatogenesis. The testicular seminiferous tubules of rats were graded according to the presence or absence of various germ cell types, including spermatozoa, spermatids, spermatocytes, spermatogonia, germ cells, and Sertoli cells, to assess histology. Each tubule was assigned a score ranging from 1 to 10. A lower Johnsen score denotes more severe dysfunction, while a higher score suggests a healthier state of spermatogenesis [86]. Tubules with a score of 1 indicate that no epithelial maturation is considered, whereas tubules with a score of 10 indicate that entire epithelial maturation is taken into consideration for the tubules with the highest activity (Table 8).

### 4.12. Statistical Analysis

A one-way analysis of variance (ANOVA) was used to assess the significant differences. Results were analyzed using Graph Pad Prism software (San Diego, CA, USA) (https://www.graphpad.com/). Tukey’s test was utilized for multiple comparisons, and *p* < 0.05 was determined to be statistically significant.

## 5. Conclusions

Collectively, ASFE treatment led to a significant amelioration of testicular tissue injury promoted by lead in rats by improving reproductive hormones, oxidative stress parameters, inflammatory mediators, and histopathological investigations. Interestingly, ASFE inhibits the production of the inflammatory cytokine IL-6 and modulates the JAK-1/STAT-3/SOCS-1 signaling pathway in male rats. However, more studies would be needed for sufficient scientific evidence of these potentials, and further investigations of the potential mitigative effects of ASFE on heavy metal toxicity and its mechanisms are required. Also, further well-designed animal and human studies should be conducted to shed light on lead toxicity and male reproduction.

## Figures and Tables

**Figure 1 ijms-25-05562-f001:**
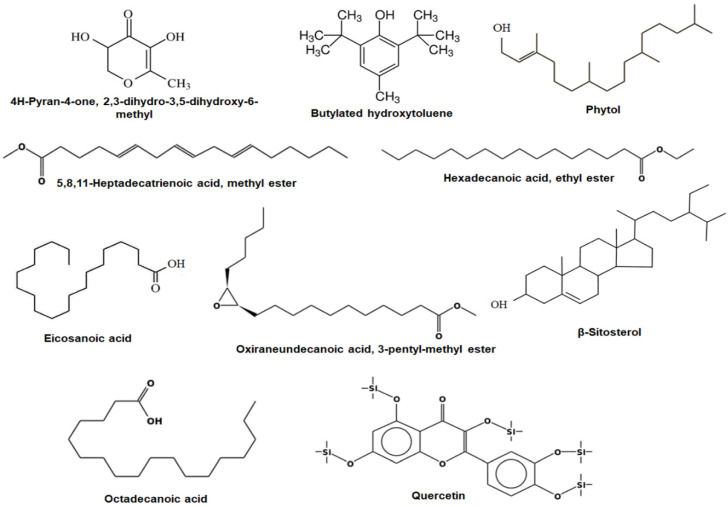
The most abundant phytochemicals in ASFE.

**Figure 2 ijms-25-05562-f002:**
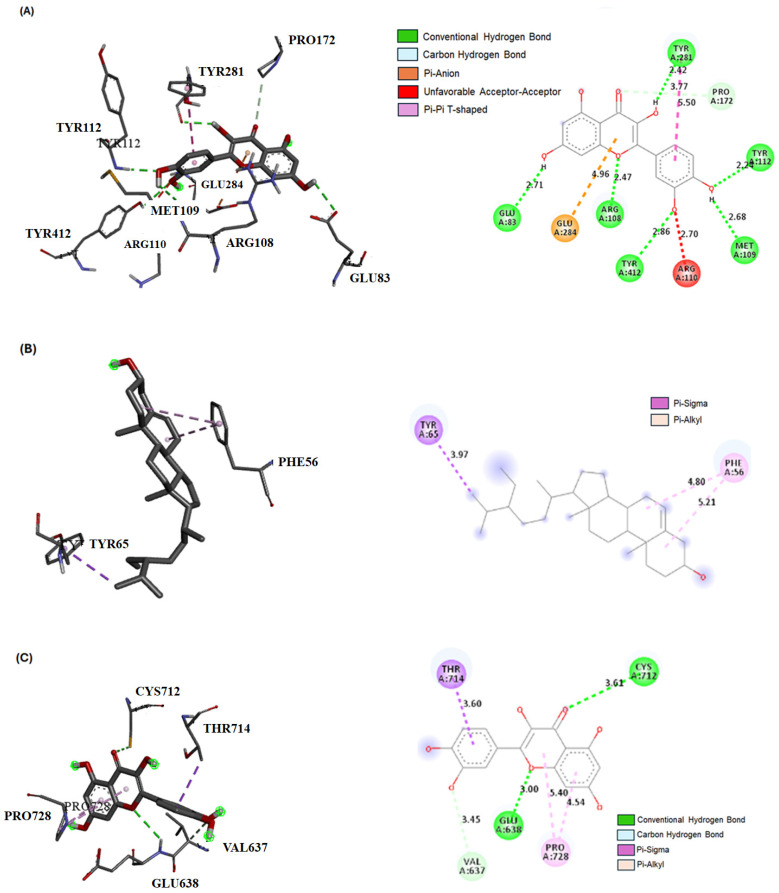
Best compounds’ interactions with JAK-1, STAT-3, and SOCS-1 proteins (2D) and (3D). (**A**) Quercetin derivative with JAK-1 (−9.1). The conventional hydrogen bond (GLU83-2.71A), conventional hydrogen bond (ARG108-2.47A), conventional hydrogen bond (TYR412-2.86A), conventional hydrogen bond (MET109-2.68A), conventional hydrogen bond (TYR112-2.24A), conventional hydrogen bond (TYR281-2.42A), pi–pi T-shaped (TY) TYR281-5.50A), pi–anion (GLU284-4.96A), carbon–hydrogen bond (PRO172-3.77A). (**B**) β-sitosterol with SOCS-1 (−7.1). Pi–alkyl (PHE56-4.80, -5.21A), pi–sigma (THR65-3.97A). (**C**) Quercetin derivative with STAT-3 (−7.9). Conventional hydrogen bond (CYS712-3.41A), conventional hydrogen bond (GLU638-3.00A), carbon–hydrogen bond (VAL637-3.45A), pi–alkyl (PRO728-4.54, -5.40A), pi–sigma (THR714-3.60A).

**Figure 3 ijms-25-05562-f003:**
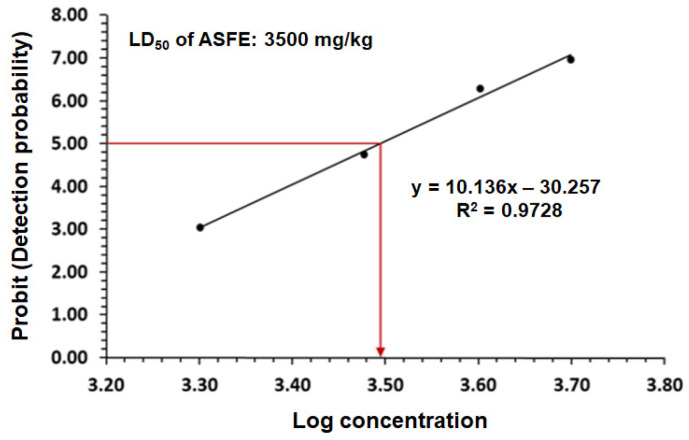
The median lethal dose (LD_50_) of ASFE.

**Figure 4 ijms-25-05562-f004:**
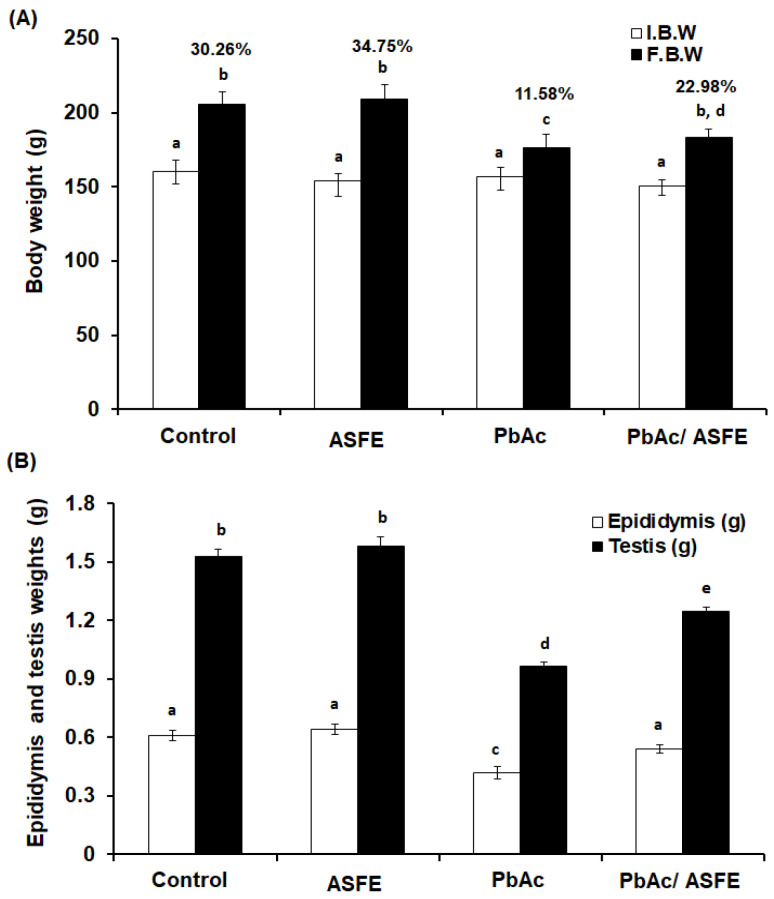
(**A**) The initial and final body weights and the percentages of the body weight changes. (**B**) The epididymis and testis weights in the different groups under study. I.B.W: initial body weight, F.B.W: final body weight, ASFE: *Annona squamosa* fruit extract, PbAc: lead acetate. Data are expressed as mean ± S.D. (*n* = 10). Means that do not share a letter showed significant difference (*p* < 0.05).

**Figure 5 ijms-25-05562-f005:**
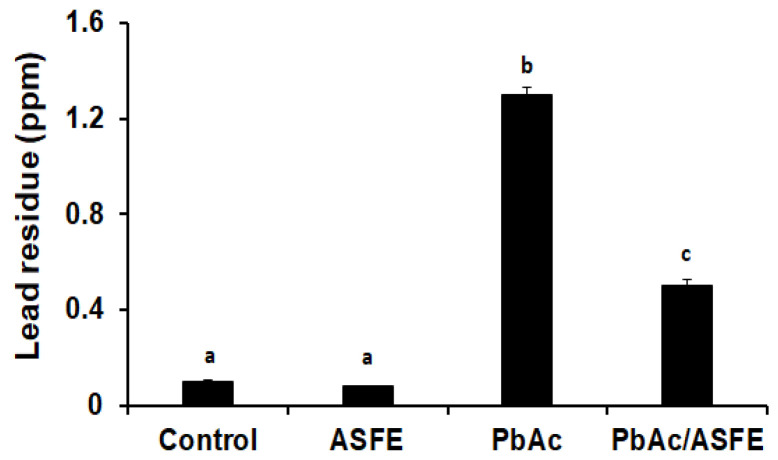
Effect of administrated PbAc plus ASFE on lead residues in the testis of male rats. ASFE: *Annona squamosa* fruit extract, PbAc: Lead acetate. Data are expressed as mean ± S.D. (*n* = 10). Means that do not share a letter showed significant difference (*p* < 0.05).

**Figure 6 ijms-25-05562-f006:**
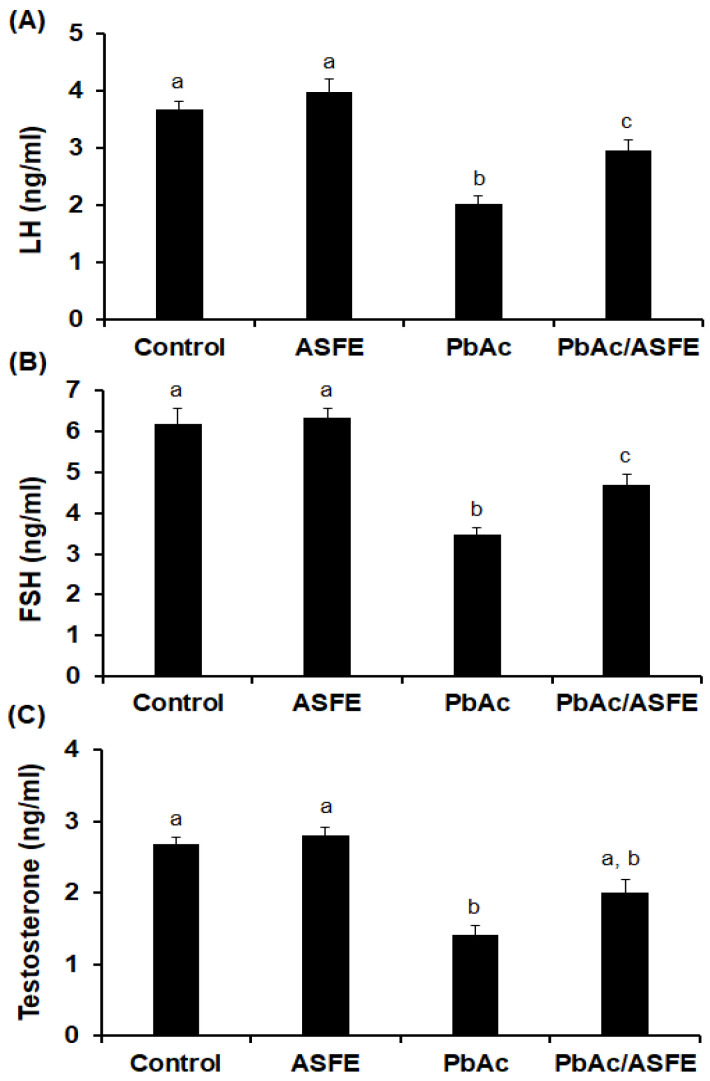
(**A**) Luteinizing hormone (LH), (**B**) follicular stimulating hormone (FSH), and (**C**) testosterone. ASFE: *Annona squamosa* fruit extract, PbAc: lead acetate. Data are expressed as mean ± S.D. (*n* = 10). Means that do not share a letter showed significant difference (*p* < 0.05).

**Figure 7 ijms-25-05562-f007:**
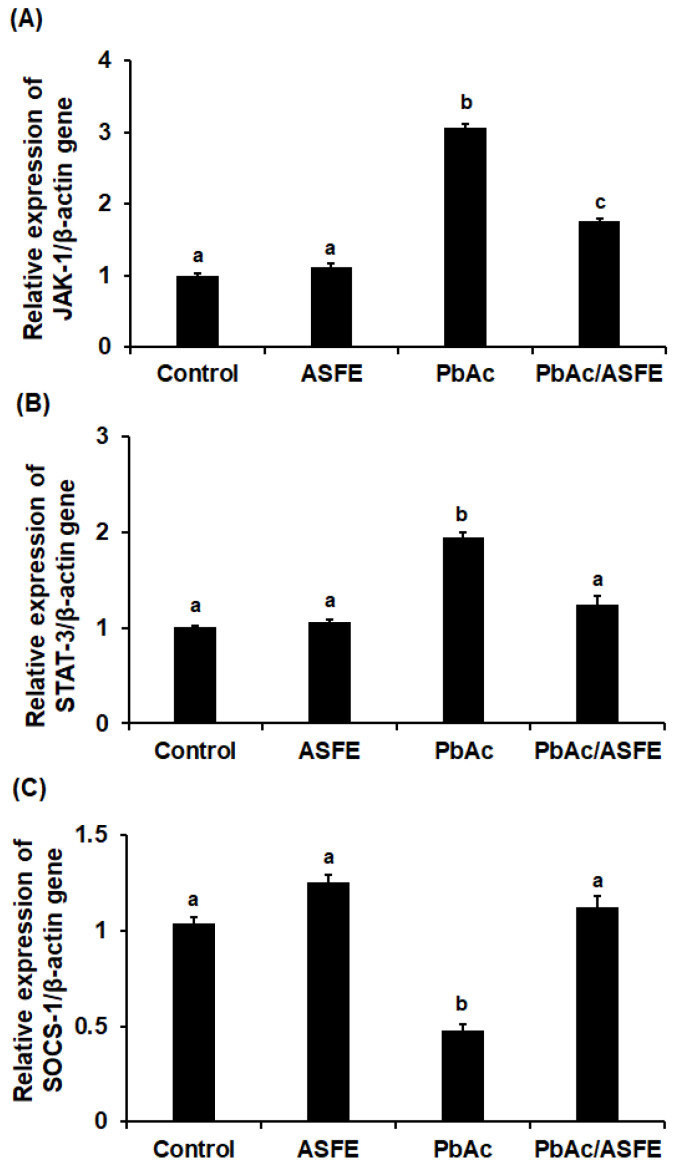
Effect of ASFE on the relative mRNA expression of *JAK-1* (**A**), STAT-3 (**B**), and *SOCS-1* (**C**) genes in lead acetate-intoxicated rats. ASFE: *Annona squamosa* fruit extract, PbAc: lead acetate, *JAK-1*: Janus kinase-1; *STAT-3*: signal transducer and activator of transcription-3; *SOCS-1*: suppressor of cytokine signaling-1. Data are expressed as mean ± S.D. (*n* = 10). Means that do not share a letter showed significant difference (*p* < 0.05).

**Figure 8 ijms-25-05562-f008:**
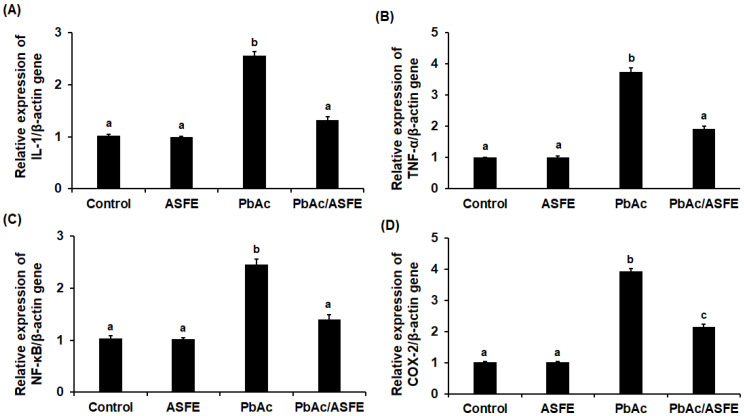
Effect of ASFE on the relative mRNA expression of *IL-6* (**A**), *TNF-α* (**B**), *NF-κB* (**C**), and *COX-2* (**D**) genes in lead acetate-intoxicated rats. ASFE: *Annona squamosa* fruit extract, PbAc: lead acetate, *IL-6*: interleukin-1, *TNF-α*: tumor necrosis factor alpha, *NF-κB*: nuclear factor kappa beta, *COX-2*: cyclooxygenase-2. Data are expressed as mean ± S.D. (*n* = 10). Means that do not share a letter showed significant difference (*p* < 0.05).

**Figure 9 ijms-25-05562-f009:**
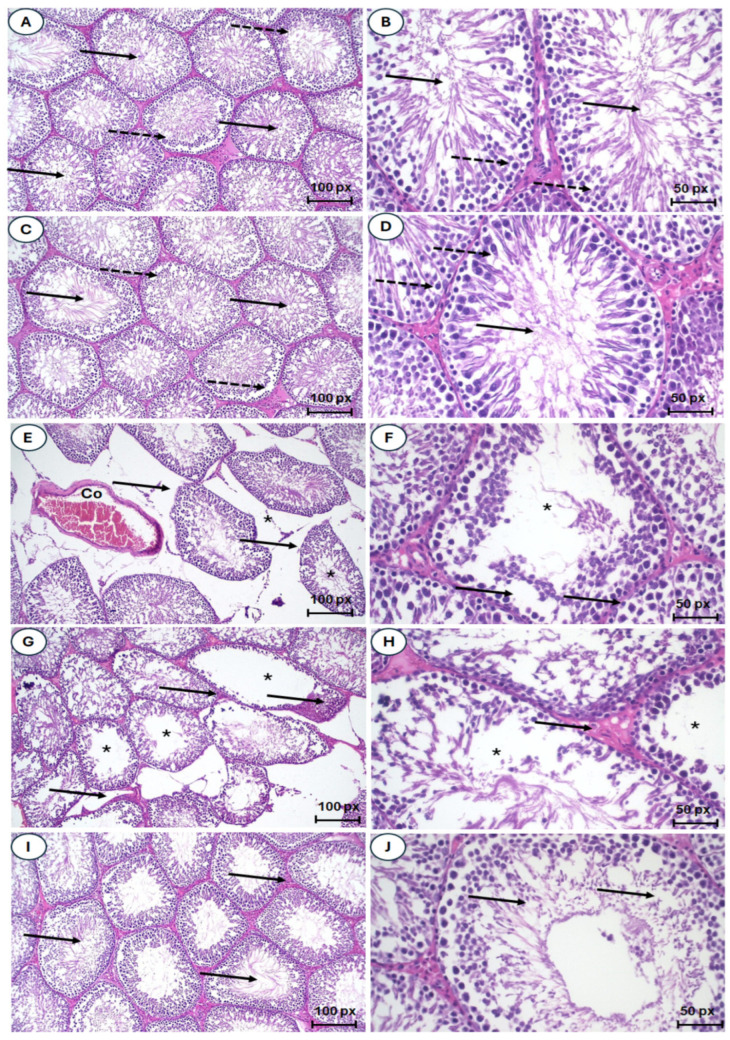
(**A**–**J**). Photomicrographs of testicular sections of the different groups. (**A**–**D**) Testis sections from the negative and ASFE control groups, which indicated normal histological structure, with normal luminal sperms (solid arrow), and spermatogonial layers (dotted arrow). (**E**–**H**) Testis sections from the PbAc-intoxicated rats with apparent morphological alteration of testicular tissue, congestion of the testicular vessels (Co), defective spermatogenic series in the seminiferous tubules (*), irregular contour of the seminiferous tubule (solid arrow), degeneration of spermatogenic epithelial series, and necrotic changes of the spermatogonial cells. (**I**,**J**) Testicular sections of PbAc-administered rats that were treated with ASFE, which had an ameliorative effect on testicular histology, restoring the architecture of the seminiferous tubule and spermatogonia cells’ integrity.

**Figure 10 ijms-25-05562-f010:**
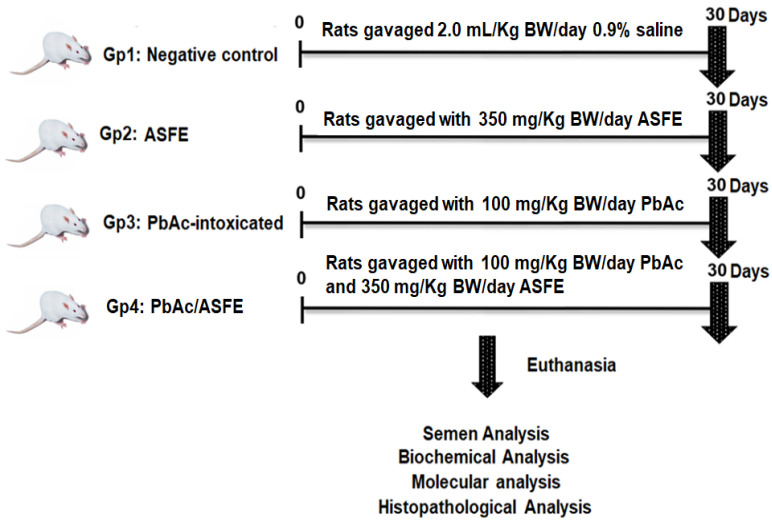
Schematic illustration of experimental design. ASFE: *Annona squamosa* fruit extract, PbAc: lead acetate.

**Table 1 ijms-25-05562-t001:** Quantitative phytochemical analysis of *A. squamosa* fruit (ASF).

Phytochemical Analysis	ASF
Metal-chelating activity (MCA) (%)	82 ± 3.87
EC_50_ of MCA (μg/mL)	383.65 ± 4.95
Total phenolic content (mg GAE/g DW)	25.48 ± 2.56
Total flavonoid contents (mg QE/g DW)	16.75 ± 1.87
Total antioxidant capacity (TAC) (mg AAE/g DW)	67.78 ± 3.58
Saponin (mg/g DW)	392 ± 3.85
Anthocyanin (mg ECG/g DW)	3.79 ± 0.29
DPPH scavenging activity (%)	86 ± 4.02
IC_50_ of DPPH (mg/mL)	5.81 ± 0.80

ASF: *Annona squamosa* fruits, MCA: metal-chelating activity, EC_50_: half-maximal effective concentration, GAE: gallic acid equivalent, QE: quercetin equivalent, DW: dry weight, TAC: total antioxidant capacity, ECG: epicatechin gallate, AAE: ascorbic acid equivalent, DPPH: diphenyl-1-picrylhydrazyl, IC_50_: half-maximal inhibitory concentration.

**Table 2 ijms-25-05562-t002:** Phytochemical constituents of ASFE that were determined by GC-MS analysis.

No.	RT (min.)	Name	MF	PA (%)
1	2.25	4H-Pyran-4-one, 2,3-dihydro-3,5-dihydroxy-6-methyl	C_6_H_8_O_4_	6.17
2	5.18	2-Methoxy-4-vinylphenol	C_9_H_10_O_2_	3.23
3	7.34	Hydroxylamine, O-decyl	C_10_H_23_NO	2.00
4	7.64	Caryophyllene	C_15_H_24_	2.72
5	8.73	Butylated hydroxytoluene	C_15_H_24_O	4.80
6	10.58	Hexadecanol	C_16_H_32_O_2_	2.35
7	11.82	Pentadecanoic acid, 14-methyl-, methyl ester	C_17_H_34_O_2_	2.14
8	17.27	9,12-Octadecadienoic acid, methyl ester	C_19_H_34_O_2_	2.66
9	18.59	5,8,11-Heptadecatrienoic acid, methyl ester	C_18_H_30_O_2_	8.61
10	19.35	Phytol	C_20_H_40_O	5.36
11	22.85	Oleic acid	C_18_H_34_O_2_	2.63
12	24.43	Eicosanoic acid	C_20_H_40_O_2_	12.51
13	25.76	Oxiraneundecanoic acid, 3-pentyl-, methyl ester	C_19_H_36_O_3_	6.42
14	27.68	β-Sitosterol	C_29_H_50_O	7.53
15	33.57	Bis(2-Ethylhexyl) phthalate	C_24_H_38_O_4_	2.19
16	39.72	Hexadecanoic acid, ethyl ester	C_18_H_36_O_2_	12.36
17	42.69	9,12,15-Octadecatrienoic acid, methyl ester	C_19_H_32_O_2_	2.65
18	43.58	Octadecanoic acid	C_18_H_36_O_2_	5.74
19	45.63	Quercetin derivative	C_30_H_50_O_7_Si_5_	7.85

ASFE: *Annona squamosa* fruit extract; RT: retention time; MF: molecular formula; PA (%): peak area percentage.

**Table 3 ijms-25-05562-t003:** The binding affinity and ∆G (Kcal/mol) for JAK-1, STAT-3, and SOCS-1 proteins with active compounds.

Compound	JAK-1	STAT-3	SOCS-1
4H-Pyran-4-one, 2,3-dihydro-3,5-dihydroxy-6-methyl	−5.8	−4.6	−4.9
2-Methoxy-4-vinylphenol	−5.4	−5.0	−5.3
Hydroxylamine, O-decyl	−4.9	−4.0	−4.4
Caryophyllene	−6.5	−6.4	−6.5
Butylated hydroxytoluene	−6.0	−5.9	−6.2
Hexadecanol	−4.2	−4.0	−3.8
Pentadecanoic acid, 14-methyl-, methyl ester	−4.8	−3.8	−4.4
9,12-Octadecadienoic acid, methyl ester	−4.9	−3.8	−4.7
5,8,11-Heptadecatrienoic acid, methyl ester	−5.0	−4.7	−4.7
Phytol	−5.1	−3.8	−4.5
Oleic acid	−5.3	−4.5	−3.9
Eicosanoic acid	−6.0	−3.8	−3.8
Oxiraneundecanoic acid, 3-pentyl-, methyl ester	−5.4	−5.1	−5.0
β-Sitosterol	−7.9	−6.8	−7.1
Bis(2-Ethylhexyl) phthalate	−6.3	−5.9	−4.8
Hexadecanoic acid, ethyl ester	−4.6	−4.0	−4.5
9,12,15-Octadecatrienoic acid, methyl ester	−5.0	−4.6	−4.5
Octadecanoic acid	−5.1	−3.9	−4.5
Quercetin derivative	−9.1	−7.9	−6.9

**Table 4 ijms-25-05562-t004:** Effect of ASFE on spermatological parameters of lead acetate-intoxicated rats.

Groups	Abnormality (%)	Count (×10^6^/mL)	Viability (%)	Motility (%)
Control	11.82 ± 1.55 ^a^	68.91 ± 3.87 ^a^	64.31 ± 4.65 ^a^	53.66 ± 4.56 ^a^
ASFE	10.21 ± 2.01 ^a^	71.24 ± 4.58 ^a^	61.53 ± 3.93 ^a^	55.79 ± 3.93 ^a^
PbAc	19.62 ± 2.59 ^b^	31.77 ± 2.79 ^b^	27.82 ± 2.95 ^b^	28.21 ± 3.15 ^b^
PbAc/ASFE	13.54 ± 2.89 ^a^	55.69 ± 3.48 ^a,c^	50.75 ± 3.86 ^a^	47.84 ± 3.79 ^a^

Data are expressed as mean ± S.D. (*n* = 10). ASFE: *Annona squamosa* fruit extract, PbAc: lead acetate; Means that do not share a letter in each column showed significant difference (*p* < 0.01).

**Table 5 ijms-25-05562-t005:** Effect of ASFE on testicular oxidant/antioxidant status of lead acetate-treated rats.

Groups	MDA (nmol/g Tissue)	PC (nmol/mg Protein)	SOD (U/mg Protein)	CAT (U/mg Protein)	GSH (mmol/g Tissue)
Control	21.73 ± 1.95 ^a^	2.80 ± 0.38 ^a^	0.67 ± 0.05 ^a^	3.88 ± 0.29 ^a^	31.21 ± 2.96 ^a^
ASFE	17.92 ± 1.45 ^a^	2.57 ± 0.26 ^a^	0.89 ± 0.07 ^a^	4.24 ± 0.38 ^a^	33.47 ± 3.12 ^a^
PbAc	44.67 ± 3.86 ^b^	6.87 ± 0.49 ^b^	0.39 ± 0.06 ^b^	1.95 ± 0.18 ^b^	17.89 ± 1.84 ^a^
PbAc/ASFE	26.84 ± 2.74 ^a^	3.72 ± 0.31 ^c^	0.58 ± 0.04 ^a^	2.87 ± 0.25 ^a,c^	24.75 ± 1.95 ^a,c^

Data are expressed as mean ± S.D. (*n* = 10). MDA: malondialdehyde; PC: protein carbonyl; SOD: superoxide dismutase; CAT: catalase; GSH: reduced glutathione; ASFE: *Annona squamosa* fruit extract, PbAc: lead acetate; Means that do not share a letter in each column showed significant difference (*p* < 0.05).

**Table 6 ijms-25-05562-t006:** Effect of ASFE on the testicular inflammatory biomarkers of lead acetate-administered rats.

Groups	IL-6 (Pg/mL/mg Protein)	TNF-α (Pg/mL/mg Protein)	NF-κB (ng/mg Tissue)	COX-2 (ng/g Tissue)
Control	28.44 ± 1.49 ^a^	58.47 ± 1.86 ^a^	18.65 ± 0.97 ^a^	225.53 ± 4.61 ^a^
ASFE	26.21 ± 1.23 ^a^	53.98 ± 1.39 ^a^	16.43 ± 1.11 ^a^	215.42 ± 5.82 ^a^
PbAc	69.87 ± 3.14 ^b^	84.29 ± 2.17 ^b^	37.69 ± 1.74 ^c^	359.86 ± 6.27 ^b^
PbAc/ASFE	34.65 ± 2.15 ^a^	64.38 ± 0.04 ^a^	27.55 ± 1.07 ^d^	302.17 ± 5.31 ^e^

Data are expressed as mean ± S.D. (*n* = 10). IL-6: interleukin 6; IL-1β: interleukin 1 beta; TNF-α: tumor necrosis factor alpha; NF-κB: nuclear factor kappa beta; COX-2: cyclooxygenase-2; ASFE: *Annona squamosa* fruit extract, PbAc: lead acetate; Means that do not share a letter in each column showed significant difference (*p* < 0.01).

**Table 7 ijms-25-05562-t007:** Forward and reverse primer sequences for RT-PCR.

Gene	Accession Number	Forward Sequence (5′–3′)	Reverse Sequence (5′–3′)
*JAK-1*	NM_053466	GAATGTACTGGGCGTCTTGG	TCAAGGAGTGGGGTTGCTTC
*STAT-3*	NM_012747	CACCCTGAAGCTGACCCAG	TATTGCTGCAGGTCGTTGGT
*SOCS-1*	NM_145879	TAACCCGGTACTCCGTGACT	CTCCCACGTGGTTCCAGAAA
*IL-6*	NM_012589	CACAAGTCCGGAGAGGAGAC	TCACAAACTCCAGGTAGAAACG
*TNF-α*	NM_012675	CTGAACTTCGGGGTGATCG	GCTTGGTGGTTTGCTACGAC
*NF-κB*	NM_001415012	CCTAGCTTTCTCTGAACTGCAAA	GGGTCAGAGGCCAATAGAGA
*COX-2*	NM_017232	CTACACCAGGGCCCTTCC	TCCAGAACTTCTTTTGAATCAGG
*β-actin*	NM_031144.3	ATCGCTGACAGGATGCAGAAG	AGAGCCACCAATCCACACAGA

*JAK-1*: Janus kinase-1; *STAT-3*: signal transducer and activator of transcription-3; *SOCS-1*: suppressor of cytokine signaling-1; *IL-6*: interleukin-6; *TNF-α*: tumor necrosis factor alpha; *NF-κB*: muclear factor kappa B; *COX-2*: cyclooxygenase 2.

**Table 8 ijms-25-05562-t008:** Histological classification of seminiferous tubular cross-sections according to the Johnsen scoring system.

Score	Description
10	Complete spermatogenesis with many spermatozoa. Germinal epithelium organized in a regular thickness leaving an open lumen.
9	Many spermatozoa are present but germinal epithelium disorganized with marked sloughing or obliteration of lumen.
8	Only a few spermatozoa are present in the section.
7	No spermatozoa but many spermatids are present.
6	No spermatozoa and only a few spermatids present.
5	No spermatozoa, no spermatids but several or many spermatocytes present.
4	Only few spermatocytes and no spermatids or spermatozoa present.
3	Spermatogonia are the only germ cells present.
2	No germ cells but Sertoli cells are present.
1	No cells in the tubular section

## Data Availability

The original contributions presented in the study are included in the article, further inquiries can be directed to the corresponding author/s.

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
