# Peer review of "Annona squamosa Fruit Extract Ameliorates Lead Acetate-Induced Testicular Injury by Modulating JAK-1/STAT-3/SOCS-1 Signaling in Male Rats"

_ijms, 2024, doi:10.3390/ijms25105562_

Round 1
Reviewer 1 Report
Comments and Suggestions for Authors
This article successfully obtained Annona squamosa fruit extract (ASFE), analyzed its phytochemical composition, and constructed a PbAc-intoxicated rat model to investigate the ameliorative effects of ASFE on lead-induced testicular injury via rat body weight changes, semen analysis, biochemical analysis, molecular analysis and histopathological analysis. Despite the positive results demonstrated by the animal experiments, the relevant mechanism of action is not clearly stated and the authors' presentation is not standardized enough. There are some points that merit to be redesigned, which make this manuscript not yet suitable for publication in its current form.
1. The article is not innovative enough. The authors mentioned that there have been many studies showing the impact of plant-derived natural products on some reproductive metrics and the improvement of lead acetate-induced testicular injury. Therefore, what are the advantages of ASFE? What is the necessity of studying ASFE?
2. At the beginning of this manuscript, the chemical components of ASFE were analyzed, the major plant components were identified, and the affinity of each component with JAK-1, STAT-3 and SOCS-1 proteins was simulated via molecular docking. Subsequently, it could be further determined which component of ASFE plays a vital role, so that the experimental results are more convincing.
3. The acute toxicity (24 h) of ASFE was determined in this manuscript, and ASFE was used continuously for 30 d in a follow-up experimental protocol. However, the authors did not demonstrate the safety of ASFE at 1/10 LD50 under long-term application.
In addition to the above major issues, there are some points that also need to be revised:
1. The content of methods involved in the abstract needs to be simplified.
2. The picture format is not uniform and needs to be refined. For example:
in figure 2, the text labeling of amino acids is not clear, the different colors of the secondary structure of protein in 3D figure are not explained, and figure A is horizontally compressed;
in figure 3, the font size of label of X and Y axis is inconsistent with other figures, the tick label of y axis is wrong;
in figure 4, the label of X axis is not aligned, the Y axis is not marked with units;
In table 1, "82%±3.87" should be "82±3.87";
3. The results in lines 273-274 are incorrectly stated.
4. The discussion part repeats a lot of experimental results, which can be simplified.
Comments on the Quality of English LanguageExtensive editing of English language required.
Author Response
Thank you for the review of our manuscript (ijms-2984218) titled “Annona squamosa fruit extract ameliorates lead acetate-induced testicular injury by modulating JAK-1/STAT-3/SOCS-1 signaling in male rats”. The whole manuscript has been revised carefully for grammar and spelling mistakes with extensive editing of English language by native. We have considered all your comments and corrected the suggested changes. These revisions are highlighted in the revised manuscript using track changes and are summarized below.

Reviewer 2 Report
Comments and Suggestions for Authors
1. Addressing lead contamination in clinical settings or everyday situations raises questions about the efficacy and feasibility of the suggested approach. Is the strategy more cost-effective or simpler to implement?
2. The Abstract should be succinct and precise, emphasizing the significant findings and core conclusions. These aspects must be underscored with clearer, more exact terminology.
3. Enhanced discussions on toxic ion elimination could benefit from references such as 10.1016/j.ijbiomac.2019.09.197 and 10.1021/acssuschemeng.9b02139. Additionally, the merits of the current study should be compared with those found in related published works.
4. The manuscript exhibits several formatting inconsistencies. For instance, the initial use of any abbreviation should be accompanied by an explanatory note, unless it refers to universally recognized terms such as DNA or RNA. A thorough review is advised to ensure proper use of abbreviations.
5. It is advisable that the manuscript includes a Figure Abstract. This concise element enhances the visibility of the article online, enabling readers to swiftly grasp the core of the presented data in a more organized manner.
Author Response
Dear reviewer 2,
Thank you for the review of our manuscript (ijms-2984218) titled “Annona squamosa fruit extract ameliorates lead acetate-induced testicular injury by modulating JAK-1/STAT-3/SOCS-1 signaling in male rats”. The whole manuscript has been revised carefully for grammar and spelling mistakes. We have considered all your comments and corrected the suggested changes.

Reviewer 3 Report
Comments and Suggestions for Authors
The present study investigated the chemical composition of ASFE and its potential in mitigating PbAc-induced testicular damage. While significant effects of ASFE were observed, several aspects need to be addressed for further improvement:
1. Within the results, after processing the analysis of the component of ASFE, the analysis of the binding affinity with JAK-1, STAT-3, and SOCS-1 was performed in Table 3. An explanation as to why an analysis of JAK-1~ SOCS-1 and cohesion is performed is also required. Otherwise, author need to study that correlation of JAK-1~SOCS-1 in PbAc induced testis damage.
2. mRNA expression data is typically presented graphically rather than in tables. Therefore, it would be more appropriate to represent mRNA expression levels using graphs rather than tables.
3. Please include the mRNA expression levels of enzymes involved in testosterone synthesis, such as Cyp17a1 and 3beta HSD etc, to provide a comprehensive analysis of the genes involved in testosterone production.
4. Along with the analysis of protein expression differences in JAK-1, STAT-3 (pSTAT-3), and SOCS-1, it is important to check for any changes in expression levels of these gene or proteins in the testicular tissue.
5. The assessment of testicular tissue using H&E staining alone may not be sufficient to evaluate germ cell or testis damage. It would be beneficial to include additional analyses such as germ cell marker staining or Tunnel assay to confirm cellular damage within the testis.
6. Despite analyzing the chemical composition of ASFE, an explanation is needed for why experiments were conducted using crude extract instead of a single compound. Providing rationale for this choice would enhance the clarity of the study methodology.
7. Including immunohistochemical staining for spermatozoa-specific markers such as acrosin and protamin1 within the testicular tissue would further corroborate the results of sperm analysis.
8. Although inflammatory markers were analyzed, there is a need for protein expression analysis related to reactive oxygen species (ROS) and genes/proteins associated with oxidative stress such as HO-1, Nrf2, and SOD. Alternatively, performing immunostaining using tissue samples could provide additional insights into the role of oxidative stress in lead-induced testicular toxicity.
(Reference: Zhao et al (PMID: 37716076) "Lead exposure induces inflammatory damage through oxidative stress-NF-κB signaling pathway.")
9. Further elucidation is needed on which specific compound(s) within the extract may have contributed to the mitigation of toxicity and inflammatory responses induced by PbAc.
(author known the single compound of extract although not entire content). This additional description would enhance the understanding of the mechanisms underlying the protective effects of the extract.
10. Author is well analyzed the chemical composition of ASFE, but biological analysis of effect of ASFE on damaged testis. If the research results and contents are supplemented with this stuff, it will be a more complete work.
Author Response
Dear reviewer 3,
Thank you for the review of our manuscript (ijms-2984218) titled “Annona squamosa fruit extract ameliorates lead acetate-induced testicular injury by modulating JAK-1/STAT-3/SOCS-1 signaling in male rats”. The whole manuscript has been revised carefully. We have considered all your comments and corrected the suggested changes.

Round 2
Reviewer 1 Report
Comments and Suggestions for Authors
This revised manuscript has been obviously improved. I recommend the manuscript to be accepted for publication in its present form.
Reviewer 3 Report
Comments and Suggestions for Authors
accepted